# Assessing the Hypertension Risk: A Deep Dive into Cereal Consumption and Cooking Methods—Insights from China

**DOI:** 10.3390/nu16173027

**Published:** 2024-09-08

**Authors:** Yingyu Huang, Yang Ao, Xuzhi Wan, Xiaohui Liu, Jianxin Yao, Hao Ye, Anli Wang, Pan Zhuang, Jingjing Jiao, Yu Zhang

**Affiliations:** 1Department of Gastroenterology, The First Affiliated Hospital, Zhejiang University School of Medicine, Hangzhou 310003, China; yingyuhuang@zju.edu.cn (Y.H.); xuzhiwan@zju.edu.cn (X.W.); yaojianxin@zju.edu.cn (J.Y.); anliwang@zju.edu.cn (A.W.); panzhuang@zju.edu.cn (P.Z.); 2Department of Food Science and Nutrition, College of Biosystems Engineering and Food Science, Zhejiang University, Hangzhou 310058, China; 3Department of Endocrinology, The Second Affiliated Hospital, Zhejiang University School of Medicine, Hangzhou 310009, China; aoyang9@zju.edu.cn (Y.A.); liuxiaohui@zju.edu.cn (X.L.); yh1999@zju.edu.cn (H.Y.); jingjingjiao@zju.edu.cn (J.J.); 4Department of Nutrition, School of Public Health, Zhejiang University School of Medicine, Hangzhou 310058, China

**Keywords:** cereals, refined grains, whole grains, cooking methods, hypertension

## Abstract

Background: Cereal grains are rich in carbohydrates and could trigger a hyperglycemic response which is closely linked to blood pressure status. We aim to examine the associations between the consumption of cereals with different cooking methods and hypertension risk. Methods: We conducted a prospective analysis utilizing the nationwide data of 11,080 adult participants who were free of hypertension at baseline. Cereal intake was assessed using 3-day 24 h dietary recalls with a weighing technique. Hypertension incidence was identified in adherence with the Seventh Joint National Commission guidelines during the follow-up. Cox proportional hazards regression models were used to extrapolate hazard ratios associated with hypertension risk. Results: Over an average follow-up span of 7 years (77,560 person-years), we identified 3643 new hypertension cases. The intake of total, fried, and baked cereals was associated with 15%, 20%, and 20% higher risk of hypertension, respectively. Whole grain consumers had an 8% lower risk of hypertension compared with non-consumers, while total refined grain consumers showed no significant association. Replacing one daily serving of fried or baked cereals with an equivalent serving of boiled cereals was related to a 28% or 14% lower risk, respectively. Conclusions: Total, fried, and baked cereal consumption was positively associated with hypertension risk, while consuming whole grains was related to a lower risk. Modifying cooking methods from frying or baking to boiling for cereals may be beneficial to lower risk. The current study underscores the significance of considering both the degree of processing and cooking methods applied to cereals in addressing hypertension prevention and management.

## 1. Introduction

Hypertension, one of the leading risk factors for premature death and disability worldwide, has demonstrated a rapid surge in prevalence. From 1990 to 2019, the global tally of cases diagnosed as hypertension showed a drastic augmentation, doubling from 650 million to a staggering 1.3 billion [1]. Such an escalating health concern runs parallel with urbanization, aging, and rapid economic growth, emphasizing the need for identifying modifiable risk factors to support the primary prevention of hypertension. Multiple studies have concurred that dietary factors play an integral role in the genesis and escalation of hypertension [2,3]. For instance, an excessive intake of sodium emerges as a primary, yet controllable, risk factor implicated in the incidence of hypertension. Numerous well-documented examples of healthy dietary patterns have shown their efficacy in reducing blood pressure, such as the Mediterranean diet [4,5] and the dietary approaches to stop hypertension [6,7,8]. These approaches advocate for a dietary pattern abundant in fruits, vegetables, and low-fat dairy products, along with the minimization of sodium and saturated fat intake, as a viable method for preventing high blood pressure.

Cereal grains, the cornerstone of the traditional human diet [9], account for approximately 53.1% of the total calorie intake along with significant quantities of dietary fiber, carbohydrates, protein, and vitamin B2 [10]. As per the Chinese Dietary Guidelines (2022), it is recommended for adults to consume an average of 200–300 g of cereals daily, including 50–150 g of whole grains and mixed legumes [11]. Grains can be categorized as whole grains or refined grains according to their degree of processing. The health benefits of whole grains in lowering blood pressure are well documented in the findings from some prospective cohort studies in Western populations [12,13]. However, refined grain intake is frequently associated with detrimental health outcomes, including an elevated risk of CVD, T2D, and hypertension. To enhance dietary quality, the 2020–2025 US Dietary Guidelines Advisory Committee has advised limiting the consumption of refined grains and ensuring that at least half of the daily grain intake is comprised of whole grains [14]. However, a comprehensive meta-analysis of three distinct studies, encompassing 18,842 newly diagnosed cases of hypertension, revealed no discernible association between refined grain intake and the risk of hypertension. Surprisingly, augmenting refined grain consumption by 30 g/d demonstrated no association with the onset of hypertension [15]. In general, the correlation between the intake of grains with different degrees of processing and the risk of hypertension remains a contentious issue, especially as evidence is scant in non-Western populations harboring distinct patterns of grain intake.

Cooking methods profoundly influence the chemical composition, nutritional attributes, and the bioavailability of nutrients in food items, which subsequently impact blood pressure levels [16,17]. For instance, cooking methods like deep-frying or baking have been shown to augment the concentrations of unhealthy fatty acids in food—notably saturated fatty acids (SFAs) and trans fatty acids (TFAs)—which may trigger elevated cholesterol levels and thus raise blood pressure. Stir-frying or prolonged boiling at high temperatures may disrupt the beneficial nutrients in foods, such as potassium, calcium, and magnesium. Prior studies have illuminated a more pronounced correlation between weight gain and the consumption of French fries, compared to that of boiled, baked, or mashed potatoes [18], which highlights the influence of various cooking methods on the observed associations. However, limited epidemiological evidence has directly linked the consumption of cereals prepared using different cooking methods to the risk of hypertension, necessitating further studies to elucidate the associations of different processing degrees of cereals and different cooking methods with the risk of hypertension along with underlying biological mechanisms.

To address these limitations and fill identified gaps, our study aims to prospectively evaluate the correlation between the consumption of whole grains, refined grains, and total cereals prepared by four distinct cooking techniques—boiling, stir-frying, frying, and baking—and the risk of hypertension. Herein, we utilized data sourced from the China Health and Nutrition Survey (CHNS), which was conducted from 1997 through 2015. The results may offer valuable insights into formulating dietary interventions aimed at preventing and managing hypertension.

## 2. Materials and Methods

### 2.1. Study Design and Population

The CHNS is a nationally representative multipurpose prospective cohort study in China. The ongoing CHNS was established in 1989 across ten rounds of follow-up surveys in the years 1989, 1991, 1993, 1997, 2000, 2004, 2006, 2009, 2011, and 2015. Follow-up intervals varied but were typically conducted every 2 to 3 years. The study encompasses participants selected from nine geographically diverse provinces and three largely urban cities—Beijing, Shanghai, and Chongqing—portraying a collective representation of over 553 million individuals. Based on a multistage, randomized cluster method, the survey comprises more than 30,000 participants from variable socio-economic backgrounds and geographical locations. In finer detail, each province contributes two cities, typically comprising the provincial capital and a low-income city, and four economically diverse counties (low, middle, and high income). Following this, two urban and suburban communities from each city and a single community with three rural villages from each county were random selected, with each community further consisting of 20 households. The trained interviews were given to all household members. More details about the cohort design, recruitment methods, and quality control process have been described elsewhere [19,20].

The collection of information on the diagnosis of hypertension started in 1997, thus, we adopted data from 1997, 2000, 2004, 2006, 2009, 2011, and 2015 for the current analysis.

During these rounds, participants were invited back to test sites to undergo various health assessments, including blood pressure measurements and the collection of information regarding medications and diagnosis history. Among a total of 29,476 participants, we excluded persons who were lost to follow-up (*n* = 4923) or aged <20 years (*n* = 8706). Among the remaining 15,127 adults, those having diabetes, CVD, or cancer (*n* = 750), suffering from hypertension, or without blood pressure data (*n* = 3116), as well as those with extreme energy intake (<800 or >4200 kcal/day for men and <600 or >3500 kcal/day for women) (*n* = 181) at baseline were excluded. Finally, 11,080 participants were eligible for the current analysis (Appendix A). The methodology employed was duly endorsed by the Institutional Review Committees of the University of North Carolina at Chapel Hill and the National Institute for Nutrition and Health affiliated with the Chinese Center for Disease Control and Prevention. All participants of the CHNS offered their written informed consent.

### 2.2. Assessment of Hypertension

In adherence with the Seventh Joint National Commission guidelines [21,22], blood pressure measurement procedure was standardized as follows. After a ten-minute seated rest period, qualified physicians employed a mercury sphygmomanometer and an appropriate-sized cuff to measure systolic (SBP) and diastolic blood pressure (DBP). The measurement was repeated 3 times with a 30 s interval to obtain the average value. Hypertension was defined as an SBP ≥ 140 mmHg or DBP ≥ 90 mmHg, ongoing treatment with antihypertensive medication, or a pre-existing hypertension diagnosis. Participants who received a hypertension diagnosis following the baseline survey were considered new cases in our prospective analysis.

### 2.3. Assessment of Cereal Consumption

Dietary assessment in CHNS was carried out from two perspectives, including the individual and household. The individual diet was evaluated by a 3-day 24 h dietary recall using questionnaires. These recalls were conducted over three consecutive days (2 weekdays and 1 weekend day) for each participant. Trained interviewers asked participants to recall and report all foods and beverages consumed in the previous 24 h, capturing detailed information on the type, quantity, and preparation methods of the foods consumed. The household diet was assessed using a food-weighing method, where trained technicians weighed and documented the household inventory for the same three days, including foods purchased, foods consumed, and foods remaining at the end of each day. The weighing process was meticulous, with technicians using calibrated scales to ensure precise measurements. For each round, matching versions of the Chinese Food Composition Table (FCT) [23] were utilized to determine nutrient intakes and total energy. The accuracy of 24 h dietary recall, employed to measure energy and nutrient intake, was validated [24,25]. The dietary evaluations were meticulously conducted by adept technicians from the National Institute of Nutrition and Food Safety, a branch of the Chinese Center for Disease Control and Prevention [19,20]. Based on the collected data, we examined the frequency and volume of daily average consumption of total cereals, refined grains, and whole grains across four distinct cooking methods, including boiling, frying, stir-frying, and baking. In our report, the terms “grains” and “cereals” were used with specific nuances. “Cereals” consistently highlighted specific types of grain consumption, particularly those that are staples in the Chinese diet. However, “grains” may be referenced when discussing dietary guidelines and conceptual frameworks for understanding carbohydrate intake and health outcomes. Our intent was to ensure clarity, especially regarding health implications related to different cooking methods and processing levels of these foods. To attenuate within-individual variation, cumulative average consumption was embraced. Total energy intake, total sodium intake, total potassium intake, and the primary cereal products within each cooking method group were evaluated through the Chinese FCT [23]. We utilized the Alternative Healthy Eating Index-2010 (AHEI-2010), an improved index upon the original AHEI [26], which was constructed after an exhaustive review of the relevant literature, focusing on foods and nutrients consistently associated with reduced risk of chronic diseases [27]. The comprehensive AHEI-2010 score spans from 0 to 110, comprising 11 factors. The scoring system encourages higher consumption of six components, including vegetables, fruits, whole grains, nuts and legumes, long-chain omega-3 fatty acids, and polyunsaturated fatty acids. Conversely, it advises limited intake of four components, namely sugar-added beverages, fruit juice, processed and red meat, trans fatty acids (TFAs), and sodium. Furthermore, it advocates for moderate alcohol consumption. Each dietary factor is scored from 0, representing the least healthy eating behavior, to 10, signifying optimal healthy eating behavior.

### 2.4. Assessment of Covariates

We further recorded demographic information as well as lifestyle factors, including age (years), sex, nationality (Han or non-Han), education attainment (less than, equal to, or more than high school), marital status (single, married, or other), dwelling (urban or rural), family income (quintile), athletic activity (none, low to moderate activity, or vigorous activity), status of smoking (never, former, current, or unknown), and status of alcohol consumption (abstainer or drinker) from participants during the baseline survey. The participants’ body mass index (BMI) was computed by dividing their weight in kilograms by the square of their height in meters (kg/m^2^).

### 2.5. Statistical Analysis

Fried cereals, baked cereals, boiled cereals, and stir-fried cereals were individually investigated for their association with the risk of hypertension after daily consumption in CHNS. The intake of all types of cereals were divided by daily calorie consumption (g/2000 kcal/day) according to the nutrient density approach [28]. Total cereal and refined grain intakes were categorized into quartiles (Q1–Q4), while whole grain intake was dichotomously split into consumers and non-consumers due to the significant portion of non-consumers and overall low consumption. The first category was served as a reference for analysis.

Individuals’ person-years of follow-up were computed from the round of admission to the year of hypertension diagnosis, censoring at death, or at final follow-up, whichever happened first. HRs and 95% CIs of the risk of hypertension were determined by employing time-dependent Cox proportional hazards models adjusted for relevant covariates based on prior research. The covariates taken into account for adjustment were in light of the prior study [29]. Model 1 was adjusted for sex and age. Model 2 included additional adjustments for nationality, marital status, BMI, household income, urbanization index, education, physical activity, smoking, alcohol consumption, and medical insurance. Model 3, on top of Model 2, was further adjusted for total energy intake, vegetable and fruit intake, total meat intake, and sodium and potassium intake. When assessing one type of cereal, we simultaneously adjusted for the intake of the other cereals. To decrease design effects, standard errors and variance of estimates were modified by community-level clustering. Trend tests were conducted in regression models using median values of each category as continuous variables. A likelihood ratio test was deployed to compare the model with only the linear term of individual cereal intake to the model with both cubic spline and linear terms.

In addition, we conducted various secondary analyses using the fully adjusted models. First, we evaluated the potential impact of substituting boiled cereals for baked, fried, and stir-fried cereals on hypertension risk, simulating an interventional study to examine whether the cooking methods of cereals affect the risk of subsequent hypertension [30]. Second, we used subgroup analyses with multiplicative interaction terms to examine whether the relationship between cereal consumption and hypertension risk varied by typical demographical factors such as age, sex, smoking and drinking status, BMIs, physical activity, education level, and household income. Sensitivity analyses were also performed to determine the robustness of findings after excluding participants with incident hypertension in the first two years, missing covariate data, extreme BMIs (<18.5 kg/m^2^ or >40 kg/m^2^), and after further adjustment for the AHEI score, health insurance and sugar-sweetened beverage (SSB) intake. SSBs encompass a wide range of liquid drinks, including fruit juice, carbonated soda, vitamin-enhanced water, energy drinks containing added sucrose, fruit juice concentrates, and predominantly high fructose corn syrup [31]. Research indicates that the consumption of SSBs may increase the risk of obesity, T2DM, hypertension, and all-cause mortality [32].

All statistical analyses and sample designs were carried out using the SAS software (version 9.4; SAS Institute, Cary, NC, USA) with a two-sided significance level set at *p* < 0.05.

## 3. Results

### 3.1. Characteristics of Participants

Table 1 shows the baseline characteristics of participants categorized by their daily intake levels of cereals. Individuals with a higher cereal intake exhibited a greater likelihood of being male, current smokers, residing in rural areas, and having a lower household income, lower energy and potassium intake, and lower education levels. However, they engaged in more physical exercise and achieved higher scores on the Alternate Healthy Eating Index (AHEI). Refined grains emerged as the most commonly consumed grain type, with boiling being the most prevalent method of cooking. The participants in the highest tertile of consumption exhibited a mean intake of 587.1 ± 5.6 g/2000 kcal/day for refined grains and 483.8 ± 2.4 g/2000 kcal/day for boiled cereals, respectively.

### 3.2. Cereal Consumption and Risk of Hypertension

Throughout a median follow-up duration of seven years (77,560 person-years), we detected 3643 (32.88%) new cases of hypertension. According to our fully adjusted model, individuals within the top quartile for total, fried, baked, and boiled cereal intake showed a 15%, 20%, 20%, and 11% higher risk of hypertension, respectively, in contrast to those in the lowest quartile (Table 2). Multivariable HRs for the risk of hypertension were 1.15 (95% CIs, 1.03–1.29; *p*-trend = 0.015) for total cereals, 1.20 (95% CIs, 1.06–1.36; *p*-trend = 0.002) for fried cereals, 1.20 (95% CIs, 1.06–1.36; *p*-trend < 0.001) for baked cereals, and 1.11 (95% CIs, 0.99–1.25; *p*-trend = 0.034) for boiled cereals. Appendix A categorizes the primary cereal products within each cooking method group.

We next investigated the association of the intake of refined grains with hypertension risk (Appendix A). Although the total intake of refined grains showed no significant association with the risk, the consumption of fried, baked, and boiled refined grains was all substantiated to increase the risk in our multivariable-adjusted model (*p*-trend = 0.006, <0.001, and 0.009, respectively). As shown in Table 3, individuals consuming whole grains had a risk of hypertension reduced by 8% compared to non-consumers. Furthermore, dose–response relationships established by restricted cubic-spline regression for different methods of cereal preparation mirrored those inverse associations from our quartile analysis (Figure 1).

### 3.3. Substitution Analysis

The substitution analysis demonstrated a notable beneficial role of boiled cereal consumption in hypertension risk (Figure 2). The replacement of one daily serving of fried or baked cereals with boiled cereals was correlated with 28% and 14% lower risk of hypertension, respectively. However, the substitution of boiled cereals for an equivalent serving of stir-fried cereals did not significantly change the associations.

### 3.4. Subgroup and Sensitivity Analyses

Subgroup analyses showed significant interaction effects between sex and total or boiled cereal consumption on hypertension risk (*p*-interaction = 0.012 and 0.002, respectively) (Appendix A), indicating a higher propensity for hypertension among female consumers of total or boiled cereals. Similarly, we also observed the associations of both total and boiled cereal consumption with the risk of hypertension were significantly stratified by smoking status (*p*-interaction = 0.001 and *p*-interaction < 0.001, respectively), non-smokers who consumed total or boiled cereals exhibited a higher risk of hypertension. A significant interaction was also discerned between boiled cereal consumption and alcohol drinking on hypertension risk (*p*-interaction = 0.029). Abstainers who consumed boiled cereal demonstrated an increased risk of hypertension. In addition, individuals under the age of 60 showed a higher tendency to suffer from hypertension among fried cereal consumers (*p*-interaction = 0.030).

As depicted in Appendix A, sensitivity analyses revealed the observed associations, particularly those between total, fried, and baked cereal consumption and hypertension risk, predominantly remained stable after further adjustment for health insurance, the AHEI-2010 score or SSB intake, or after excluding hypertension cases reported in the initial two years of follow-up, participants with incomplete covariate data, or individuals with an extreme BMI (<18.5 kg/m^2^ or >40 kg/m^2^).

## 4. Discussion

This prospective study investigated the correlations between the intake of cereals processed by different cooking methods and hypertension risk among Chinese adults. Our findings denoted that an augmented intake of total cereals, including fried and baked varieties, culminated in a significant escalation in hypertension risk. This positive association was also noted with fried, baked, and boiled refined grains. Conversely, whole grain consumers were faced with a lower hypertension risk compared to non-consumers. Crucially, our study indicated a reduction in hypertension risk when boiled cereals were utilized in place of fried or baked options, thereby underscoring an effective dietary recommendation to potentially minimize hypertension risk, though the risk may still persist.

Several studies have reported that dietary intake of cereals enriched with phytochemical compounds is potentially beneficial to the prevention of chronic diseases such as hypertension [33,34,35]. However, our current study revealed the highest tertile of total cereal consumption, compared to the lowest, was correlated with an increased risk of hypertension. We postulate that this discrepancy from previous studies may stem from the variations in cereal processing and cooking methods. The Chinese diet, characterized by a considerable intake of refined grains such as white rice, noodles, steamed buns, and baked pancakes, often has a high glycemic index (GI) (≥70) and elevates postprandial glycemic responses, which has been linked to the risk of diabetes, obesity, hypertension, cardiovascular diseases, and other health complications [36,37,38,39,40]. Steamed wheat bread typically made from refined wheat flour has a high glycemic index (GI = 88) and glycemic load (GL = 13) [41]. The rapid glucose spikes from high-GI foods can induce insulin resistance, oxidative stress, inflammation, and sympathetic nervous system activation, all of which are risk factors for hypertension [42,43]. A previous study showed that the higher consumption of white rice (GI = 83, GL = 37) is significantly associated with an increased risk of type 2 diabetes, particularly in Asian populations (Chinese and Japanese) [44]. In addition, high insulin levels can promote sodium retention, exacerbating blood pressure [43]. A study on Sydney adolescents indicated the detrimental effect of excessive carbohydrate intake on blood pressure, principally from high-GI/glycemic load (GL) foods [40]. Moreover, a systematic review and meta-analysis highlighted that the regular consumption of low-GI and low-GL meals could help maintain healthy blood pressure levels in adults [38]. Low glycemic index foxtail millet biscuits (GI = 50.8) modestly improved long-term glycemic and lipidemic control in type 2 diabetics [45]. In addition, daily consumption of 50 g of whole foxtail millet significantly reduced blood pressure (SBP/DBP by 4.13/3.49 mmHg) in untreated mildly hypertensive subjects [46]. Oats are recognized in several countries for their proven ability to lower blood cholesterol levels and postprandial glycemic response, often categorized as low-GI foods (oat bar, GI = 45, GL = 9) [41,47]. In a randomized controlled trial, supplementation with 30 g of oat bran reduced systolic blood pressure by 14.0 ± 15.5 mm Hg and diastolic blood pressure by 11.1 ± 14.6 mm Hg in 50 patients with stage 1 hypertension [48]. Thus, the GI may serve as a significant risk factor influencing the analysis of the correlation between cereal intake and hypertension risk in this study. Indeed, various investigations for the correlation between refined grain consumption and hypertension risk have yielded inconsistent results. Specifically, an NHS follow-up study found a significant positive association of specific refined grain foods with systolic blood pressure (SBP), while revealing a negative correlation with diastolic blood pressure (DBP) [49]. Complementing this, an Iranian cross-sectional study proposed that the regular consumption of refined grain products, especially noodles, could potentially augment the risk of hypertension by up to 69% [50]. Despite these findings, 11 meta-analyses of prospective cohort studies strongly concluded no significant association between refined grain consumption and hypertension risk [51], which is consistent with our study. This study also indicated that the definition of refined grain intake may confound the interpretation of findings [51].

Consuming whole grains has been proven to reduce the likelihood of hypertension development [34,52]. In our study, despite the low consumption of whole grains among Chinese adults, we still observed a significant reduction in hypertension risk, indicating whole grain consumers had an 8% lower risk than non-consumers. Our findings align with a cross-sectional study involving 435,907 Chinese participants, which linked amplified coarse grain consumption with both diminished blood pressure and reduced risk of hypertension [53]. Analogously, a three-year longitudinal analysis of 944 employed Japanese adults substantiated the correlation between regular consumption of whole grains and a decreased propensity for hypertension incidence [54]. Similar outcomes have been validated in Western populations [13,55,56,57]. Despite these insights, we emphasize the need for further research to corroborate our findings, particularly considering the low consumption rate of whole grains observed in the enrolled participants. Whole grains, unlike their refined counterparts that hold only the starchy endosperm, encompass the bran and germ components, which render a diverse nutrient profile, inclusive of anthocyanins, flavonoids, phenolic acids, and fibers. These bioactive compounds are acclaimed for their ability to prevent hypertension or regulate blood pressure levels, which are believed to operate via modulating the renin–angiotensin–aldosterone system, inflammation, and oxidative stress [34,52,58].

Notably, in our investigation, the consumption of fried and baked total cereals, as well as fried and baked refined grains, were all found to be positively linked with hypertension incidence, which has been supported by previous studies. A study in Kenya revealed a rapid rise in the incidence of circulatory disease/hypertension and diabetes, which may be attributed to the transition from traditional cooking methods such as boiling to high-temperature cooking techniques like stir-frying, deep-frying, and pan-frying for the thermal processing of wheat-based staple foods [59]. Cooking methods such as frying and baking can expose cereals to temperatures reaching up to 200 °C, whereas boiling is limited to 100 °C. These high-temperature processes can lead to significant atmospheric oxidation, which degrades essential nutrients and bioactive components, including vitamins, phenolics, and anthocyanins, thereby diminishing their health-promoting properties [60,61]. Additionally, due to diverse nutrient ingredients present in cereals, reducing sugars and lipids can react with amino acids or proteins via a series of thermal reactions during the frying and baking process, like the Maillard reaction [62,63]. Furthermore, many physicochemical changes and reactions occur during thermal processing, including oxidation, hydrolysis, pyrolysis, isomerization, and polymerization. Ultimately, these thermal reactions produce a series of processing contaminants, including TFAs [64,65], advanced glycation/lipoxidation end products, acrylamide, and oxidized triglyceride monomer, which may be detrimental to blood pressure [64,66,67,68]. In addition, external stressors significantly influence dietary choices and health outcomes [69]. A study from Italy revealed that the increasing stress and emotional distress during the COVID-19 pandemic led to unhealthy eating habits, including the high consumption of snacks and fast food, which are rich in fats, sugars, and calories [70]. This shift in diet may result in weight gain and other adverse health effects. Similarly, our study found that the consumption of fried and baked cereals, which are rich in fats and calories, was associated with a higher risk of hypertension. These findings underscore the global significance of the interplay between diet, stress, and hypertension, and emphasize the necessity of accounting for external stressors and lifestyle modifications when formulating dietary guidelines for the prevention and management of hypertension.

The frying process leads to a significant generation of TFAs, which raise LDL cholesterol levels while lowering HDL cholesterol levels. The elevated LDL cholesterol ultimately raises blood pressure by promoting the expression of the angiotensin II type 1 receptor [71,72]. A randomized controlled trial reported that the higher intake of TFAs and SFAs was linked to a higher risk of hypertension among middle-aged and elderly women [73]. Moreover, food additives such as baking powder and soda in baked cereal products are rich in sodium and excessive sodium intake is a major contributor to the occurrence of hypertension and CVD. This is corroborated by a study conducted in Turkey, which established a correlation between the consumption of high-sodium bread and an increased incidence of hypertension [74,75]. In contrast, boiling is an accepted processing method for the thermal treatment of healthy food. Transitioning cooking methods from frying to boiling has been shown to offer advantages in controlling blood pressure and obesity among the general population [76,77]. Here, our substitution analysis demonstrated that substituting boiled cereals for fried or baked cereals was associated with a lower risk of hypertension, further confirming the benefits of boiling cereals over frying and baking for promoting healthy blood pressure.

Our current study seems pioneering for the thorough examination of the complex relationship between the consumption of cereals processed by different cooking methods and the risk of hypertension among Chinese population in the CHNS. These findings offer valuable insights into potential dietary strategies for hypertension prevention and management in China. In detail, consumers may prefer shifting from frying or baking cereals to boiling them and increase the intake of whole grains such as brown rice, whole wheat, barley, oats, and quinoa. Healthcare providers may emphasize the benefits of whole grains and boiled cereals over fried or baked food. Personalized nutritional plans are suggested to incorporate whole grains and prioritize boiling as the preferred cooking method. Patients, particularly those consuming high amounts of cereals prepared through frying or baking, should regularly monitor their blood pressure. Public health policies may preferentially promote the education on the health benefits of whole grains and offer practical advice on incorporating them into meals, and encourage the adoption of healthy cooking techniques, such as boiling, rather than frying and baking. Meanwhile, more efforts may focus on formulating the dietary guidelines that advocate for the inclusion of whole grains in daily diets. To facilitate these shifts in cooking and dietary habits, public health initiatives are suggested to organize professional workshops and generalize these regimens to the public. Furthermore, continuous monitoring systems should be advanced to track updated dietary patterns in relation to hypertension prevalence, enabling data-driven policy decisions. Benefitting from an expansive dataset, the current analysis permits comprehensive adjustment for numerous potential confounding factors during model formulation and estimation. Despite these strengths, several limitations should still be noted. First, the diverse array of cereals distributed and consumed in China, all with their own distinct glycemic indexes, implies a potential discrepancy in the observed associations. Second, our study did not incorporate confounding factors such as electrolytes and energy drinks which, albeit rare in the study’s population, could potentially contribute to the dietary risk factors for developing hypertension. Third, we did not evaluate the influence of micronutrient intake—encompassing key elements like calcium, magnesium, zinc, selenium, and vitamins—on the development of hypertension. In addition, our study does not identify foods cooked at home versus those purchased away from home and the potential confounding effect of accompanying entrées. Furthermore, potential changes in dietary patterns during the follow-up period, which could introduce bias, were not accounted for. Last, causality between cereal consumption and hypertension prevalence may not be established because of the observational nature of this study.

## 5. Conclusions

In this prospective Chinese cohort study, we identified significant positive correlation between the elevated consumption of cereals, particularly total, baked, or fried ones using different cooking methods, and the risk of hypertension among the Chinese nationwide population. Contrary to the insignificant correlation observed between the consumption of total refined grains and hypertension risk, our findings indicate that an augmented intake of whole grains effectively mitigates this risk by 8%. Interestingly, our findings propose that a shift in cooking methods from frying or baking to boiling may lower hypertension risk by 28% or 14%, respectively. The current study underscores the significance of considering both the degree of processing and cooking methods applied to cereals in addressing hypertension prevention and management, aiming to tailor cereal dietary guidelines for the prevention and management of hypertension. Future research should continue to focus on the correlation between cereals and hypertension, incorporating factors such as the GI and elucidating the causal relationship.

## Figures and Tables

**Figure 1 nutrients-16-03027-f001:**
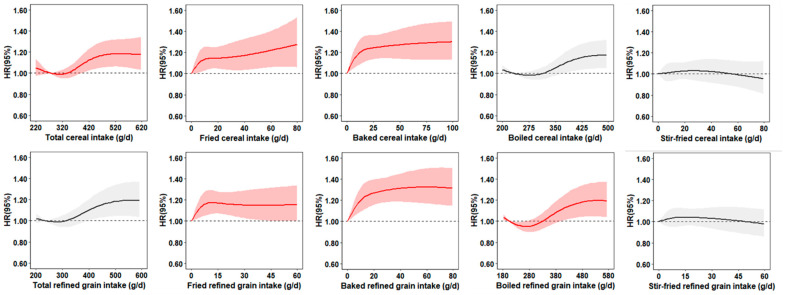
Dose–response relationships between the consumption of cereals prepared with different cooking methods and hypertension risk. Hazard ratios were estimated by restricted-cubic-spline regression adjusted for age, sex, nationality (Han or non-Han), marital status (never married, married or living as married, widowed/divorced/separated, or unknown), BMI (in kg/m^2^; <18.5, 18.5–23.9, 24–27.9, or ≥28), household income (quintile), urbanization index, education (less than high school, high school, some college or at least college), physical activity (no regular activity, low to moderate activity or vigorous activity), smoking (never, former, current, or unknown), alcohol drinking status (abstainer or drinker) and medical insurance, and total energy intake, vegetable intake, fruit intake, total meat intake, sodium intake, and potassium intake. Shaded areas represent 95% confidence intervals.

**Figure 2 nutrients-16-03027-f002:**
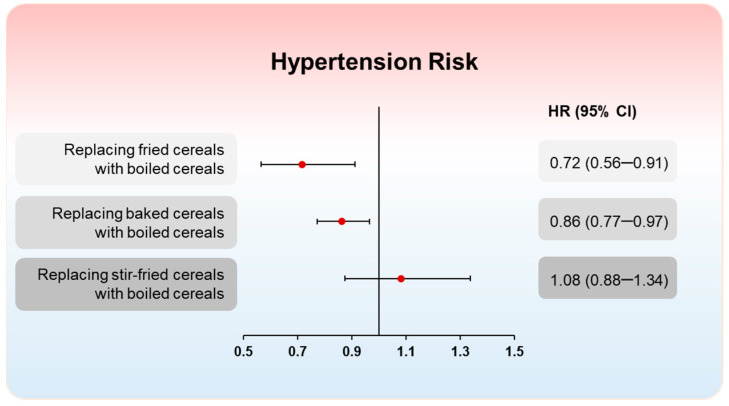
Hazard ratios (95% CIs) of hypertension risk by hypothetical substitution of one serving of boiled cereals for equal serving of fried cereals, baked cereals, and stir-fried cereals among Chinese adults. HRs were adjusted for age, sex, nationality (Han or non-Han), marital status (never married, married or living as married, widowed/divorced/separated, or unknown), BMI (in kg/m^2^; <18.5, 18.5–23.9, 24–27.9, or ≥28), household income (quintile), urbanization index, education (less than high school, high school, some college or at least college), physical activity (no regular activity, low to moderate activity, or vigorous activity), smoking (never, former, current, or unknown), alcohol drinking status (abstainer or drinker) and medical insurance, and total energy intake, vegetable intake, fruit intake, total meat intake, sodium intake, and potassium intake.

**Table 1 nutrients-16-03027-t001:** Baseline characteristics of participants according to the consumption of total cereals in CHNS 1997–2015 *.

Characteristics	Dietary Cereal Intake (g/2000 kcal/day)			
	Q1	Q2	Q3	Q4
N	2770	2770	2770	2770
Age (years)	41.7 ± 0.3	40.7 ± 0.3	39.8 ± 0.2	39.9 ± 0.2
Body mass index (kg/m^2^)	22.7 ± 0.1	22.5 ± 0.1	22.3 ± 0.1	22.4 ± 0.1
Household income (CNY/yr) ^†^	42,041.1 ± 960.2	29,604.7 ± 712.1	24,881.6 ± 541.6	23,020.4 ± 510.2
Urbanization index	72.7 ± 0.3	62.6 ± 0.4	56.6 ± 0.4	51.5 ± 0.4
Male (%)	41.9	44.0	46.1	46.3
Han (%)	90.9	86.6	86.4	88.6
Married (%)	84.7	84.4	86.0	85.1
Greater than high school (%)	14.8	6.0	4.6	3.9
Moderate-to-vigorous activity (%)	29.1	47.2	59.4	66.4
Current drinker (%)	36.4	32.3	34.5	31.8
Current smoker (%)	27.1	28.0	29.2	29.8
Dietary intake				
Energy intake (kcal/day)	2160.4 ± 9.9	2151.8 ± 8.6	2119.1 ± 9.0	2005.3 ± 10.4
Whole grains (g/2000 kcal/day)	3.4 ± 0.2	3.9 ± 0.2	5.3 ± 0.3	9.9 ± 0.8
Refined grains (g/2000 kcal/day)	257.8 ± 1.6	338.0 ± 1.6	403.1 ± 2.1	587.1 ± 5.6
Fried cereals (g/2000 kcal/day)	5.4 ± 0.2	6.2 ± 0.3	6.8 ± 0.3	8.0 ± 0.5
Stir-fried cereals (g/2000 kcal/day)	7.4 ± 0.3	8.4 ± 0.3	10.8 ± 0.4	12.6 ± 0.6
Boiled cereals (g/2000 kcal/day)	237.3 ± 1.1	316.4 ± 0.8	366.8 ± 0.9	483.8 ± 2.4
Baked cereals (g/2000 kcal/day)	7.5 ± 0.4	8.3 ± 0.4	13.0 ± 0.6	19.0 ± 0.9
Sodium (mg/2000 kcal/day)	5374.3 ± 458.4	5043.1 ± 375.2	4644.0 ± 239.5	4954.2 ± 413.0
Potassium (mg/2000 kcal/day)	1972.9 ± 22.4	1799.6 ± 14.1	1770.1 ± 13.9	1736.5 ± 14.4
AHEI score ^‡^	42.5 ± 0.1	42.9 ± 0.1	44.5 ± 0.1	44.9 ± 0.1

* Data are means ± SE unless otherwise indicated. ^†^ Household income: inflated to 2009 of the RMB. ^‡^ AHEI score spans from 0 to 110, comprising 11 factors. The scoring system encourages higher consumption of six components, including vegetables, fruits, whole grains, nuts and legumes, long-chain omega-3 fats, and polyunsaturated fatty acids. Conversely, it advises limited intake of four components, namely sugar-added beverages, fruit juice, processed and red meat, trans fats, and sodium. Furthermore, it advocates for moderate alcohol consumption. Each dietary factor is scored from 0, representing the least healthy eating behavior, to 10, signifying optimal healthy eating behavior.

**Table 2 nutrients-16-03027-t002:** Hazard ratios (95% CIs) of hypertension risk according to cereal consumption in CHNS 1997–2015.

	Quartiles of Dietary Cereal Intake (g/2000 kcal/day)				
	Q1	Q2	Q3	Q4	*p*-Trend
**Total cereals**					
Median (g/2000 kcal/day)	276	344	402	498	
Cases/person-years	755/16,620	952/24,930	948/24,930	988/19,390	
Model 1 *	1	1.01 (0.92–1.11)	1.01 (0.91–1.11)	1.12 (1.02–1.23)	0.026
Model 2 ^†^	1	1.04 (0.94–1.15)	1.07 (0.97–1.18)	1.16 (1.05–1.29)	0.004
Model 3 ^‡^	1	1.05 (0.95–1.15)	1.06 (0.96–1.18)	1.15 (1.03–1.29)	0.015
**Fried cereals**					
Median (g/2000 kcal/day)	0	6	16	38	
Cases/person-years	2436/55,608	473/12,540	406/9414	328/6270	
Model 1 *	1	1.12 (1.01–1.23)	1.23 (1.10–1.36)	1.42 (1.26–1.59)	<0.001
Model 2 ^†^	1	1.06 (0.96–1.17)	1.15 (1.04–1.28)	1.25 (1.11–1.41)	<0.0001
Model 3 ^‡^	1	1.04 (0.94–1.16)	1.11 (0.99–1.24)	1.20 (1.06–1.36)	0.002
**Baked cereals**					
Median (g/2000 kcal/day)	0	9	23	70	
Cases/person-years	2391/55,349	447/11,627	430/9522	375/7406	
Model 1 *	1	1.11 (1.01–1.23)	1.43 (1.29–1.59)	1.30 (1.17–1.45)	<0.001
Model 2 ^†^	1	1.07 (0.97–1.19)	1.33 (1.20–1.47)	1.21 (1.08–1.35)	<0.001
Model 3 ^‡^	1	1.07 (0.96–1.19)	1.33 (1.19–1.49)	1.20 (1.06–1.36)	<0.001
**Boiled cereals**					
Median (g/2000 kcal/day)	235	314	369	467	
Cases/person-years	803/13,850	915/24,930	950/24,930	975/24,930	
Model 1 *	1	0.88 (0.80–0.96)	0.88 (0.80–0.97)	0.95 (0.87–1.05)	0.519
Model 2 ^†^	1	0.92 (0.83–1.01)	0.96 (0.87–1.07)	1.02 (0.92–1.13)	0.452
Model 3 ^‡^	1	0.98 (0.89–1.08)	1.04 (0.94–1.16)	1.11 (0.99–1.25)	0.034
**Stir-fried cereals**					
Median (g/2000 kcal/day)	0	7	16	40	
Cases/person-years	1987/39,810	648/17,772	586/13,338	422/10,374	
Model 1 *	1	0.94 (0.86–1.03)	1.01 (0.92–1.11)	0.93 (0.84–1.04)	0.353
Model 2 ^†^	1	0.95 (0.87–1.04)	1.04 (0.95–1.14)	0.97 (0.87–1.08)	0.863
Model 3 ^‡^	1	0.95 (0.87–1.04)	1.05 (0.96–1.16)	0.99 (0.89–1.11)	0.733

Time-dependent Cox proportional hazard regression models were used to assess HRs (95% CIs) of hypertension. CI, confidence interval; HR, hazard ratio. * model 1 adjusting for age and sex. ^†^ model 2 adjusting for model 1 plus nationality (Han or non-Han), marital status (never married, married or living as married, widowed/divorced/separated, or unknown), BMI, household income (quintile), urbanization index, education (less than high school, high school, some college or at least college), physical activity (no regular activity, low to moderate activity, or vigorous activity), smoking (never, former, current, or unknown), alcohol drinking status (abstainer or drinker) and medical insurance. ^‡^ model 3 adjusting for model 2 plus total energy intake, vegetable intake, fruit intake, total meat intake, sodium intake, and potassium intake.

**Table 3 nutrients-16-03027-t003:** Hazard ratios (95% CIs) of hypertension risk according to whole grain consumption in CHNS 1997–2015.

Whole Grain Consumer			
	No	Yes	*p* Value
**Total whole grains**			
Cases/person-years	2785/60,963	858/21,339	
Model 1 *	1	0.95 (0.88–1.02)	0.149
Model 2 ^†^	1	0.93 (0.86–1.00)	0.050
Model 3 ^‡^	1	0.92 (0.85–0.99)	0.027
**Fried whole grains**			
Cases/person-years	3622/77,154	21/522	
Model 1 *	1	0.87 (0.57–1.34)	0.524
Model 2 ^†^	1	0.84 (0.55–1.30)	0.436
Model 3 ^‡^	1	0.86 (0.56–1.32)	0.476
**Baked whole grains**			
Cases/person-years	3571/76,328	72/1232	
Model 1 *	1	1.13 (0.89–1.42)	0.324
Model 2 ^†^	1	1.04 (0.82–1.32)	0.731
Model 3 ^‡^	1	1.03 (0.81–1.31)	0.817
**Boiled whole grains**			
Cases/person-years	3023/65,933	620/14,949	
Model 1 *	1	0.98 (0.90–1.07)	0.623
Model 2 ^†^	1	0.96 (0.88–1.04)	0.310
Model 3 ^‡^	1	0.96 (0.87–1.04)	0.314
**Stir-fried whole grains**			
Cases/person-years	3423/73,017	220/5841	
Model 1 *	1	0.83 (0.72–0.95)	0.007
Model 2 ^†^	1	0.85 (0.74–0.98)	0.022
Model 3 ^‡^	1	0.85 (0.74–0.98)	0.025

Time-dependent Cox proportional hazard regression models were used to assess HRs (95% CIs) of hypertension. CI, confidence interval; HR, hazard ratio. * model 1 adjusting for age and sex. ^†^ model 2 adjusting for model 1 plus nationality (Han or non-Han), marital status (never married, married or living as married, widowed/divorced/separated, or unknown), BMI, household income (quintile), urbanization index, education (less than high school, high school, some college or at least college), physical activity (no regular activity, low to moderate activity, or vigorous activity), smoking (never, former, current, or unknown), alcohol drinking status (abstainer or drinker) and medical insurance. ^‡^ model 3 adjusting for model 2 plus total energy intake, vegetable intake, fruit intake, total meat intake, sodium intake, and potassium intake.

## Data Availability

The data described in the manuscript will be made available upon reasonable request to the corresponding authors.

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
