# Peer review of "Assessing the Hypertension Risk: A Deep Dive into Cereal Consumption and Cooking Methods—Insights from China"

_nutrients, 2024, doi:10.3390/nu16173027_

Round 1

Reviewer 1 Report

Comments and Suggestions for Authors

The main purpose of the article is to examine the associations between the consumption of cereals prepared using different cooking methods and the risk of hypertension in a Chinese population. The study aims to provide insights into how varying cereal processing and cooking techniques—such as boiling, stir-frying, frying, and baking—affect hypertension risk. The research highlights the significant influence of cooking methods on the nutritional attributes and health impacts of cereals, emphasizing the potential benefits of boiling over frying or baking in reducing hypertension risk​.

To improve the paper, the authors could consider the following suggestions:

- Include additional potential confounding factors such as electrolyte intake (e.g., calcium, magnesium, zinc), micronutrient intake (e.g., selenium, vitamins), and consumption of energy drinks.

- Include a detailed analysis of the glycemic index (GI) and glycemic load (GL) of different cereals and their correlation with hypertension risk.

- Explore the impact of different types of refined grains, considering their GI values, to better understand their association with hypertension.

- Perform more detailed subgroup analyses to explore how different demographic factors (e.g., age, gender, socioeconomic status) interact with cereal consumption and hypertension risk.

- Discuss the implications of the findings for public health policies and dietary guidelines in more detail.

- Provide actionable recommendations for consumers, healthcare providers, and policymakers based on the study results

- Extend the literature on the matter, with articles considering other populations too. For example it would be worth including this paper https://doi.org/10.23751/pn.v23i2.11755 that provides insights into how stress and changes in lifestyle during the pandemic led to unhealthy eating habits and weight gain, particularly among Italian menopausal women. This context can reinforce the idea that dietary choices, influenced by external stressors, significantly affect health outcomes such as hypertension. 

By addressing these areas, the paper could provide a more comprehensive and detailed understanding of the relationship between cereal consumption, cooking methods, and hypertension risk, ultimately contributing to more effective dietary recommendations and public health strategies.

Reviewer 2 Report

Comments and Suggestions for Authors

The aim of this study was to examine the relationship between intake of cereals with different cooking methods and the risk of hypertension. The prospective study was performed in over 11,000 Chinese adults. free of hypertension at baseline and the patients were followed-up for 7 years. The results indicate that the intake of total, fried and baked sereals was associated with greater risk of hypertrension. However, whole grain consumption was associated with lower risk of hypertension. Replacement of fried or booked cereals with whole grains decreased the risk of hypertension.

The topic and the results are of interest and the manuscript is well-written. The results provide an important insights into dietary determinants of hypertension risk. I have no critical comments.

Author Response

Comment: The topic and the results are of interest and the manuscript is well-written. The results provide an important insights into dietary determinants of hypertension risk. I have no critical comments.

Response: We appreciate the reviewer’s comment. Thank you for your positive feedback and highlighting the significance of our study.

Reviewer 3 Report

Comments and Suggestions for Authors

This prospective cohort study examined links between cereal consumption and risk for hypertension.  A novel aspect of this work was the separation of cereals by cooking method:  boiled, fried, stir-frying, an baking.  The results, that cooking methods did indeed significantly impact risk, are intriguing and potentially quite relevant. 

A table identifying the main cereal products in each of the cooking groups is needed to fully appreciate the relevance of these results.

To what degree were the cereals commercially prepared versus prepared in the home?  Since food away from home (e.g., store-bought or purchased as carry-out or consumed at restaurants) would likely have more unhealthy additives.   For example, popcorn consumed at home would likely have less added fat, salt, and sugar than if purchased at a movie theater or street fair.  Another example would be rice boiled at home versus boiled at a restaurant – the latter will likely have more add oil and salt or other spices.   

Furthermore, a discussion is needed regarding foods that typically accompany cereal/grain consumption which is a key confounding (and unknown) variable in this investigation.  Boiled rice is often consumed with high-sodium chicken dishes (e.g., chicken teriyaki).  Dumplings or egg drop soup (both high sodium) is often served with fried rice.  Pancakes are served with high sugar syrups, etc.  This is a complex issue considering that cereals and grains are typically ‘side’ dishes – and the entree may have the greater impact on health (e.g., plant based versus animal based).

2.2.  Please clarify how often participants returned to test sites where blood pressure was recorded and information collected regarding medications and diagnosis history. 

2.3.  Two different methods are described for diet assessment:  recalls and food-weighing in the home.  Clearly explain how these two diet methods were used in the present study. 

Are the terms ‘grains’ and ‘cereals’ used interchangeably in this report? 

Line 225 and 226:  total, fried, and baked cereal data are provided – but boiled is not mentioned (11% and p<0.05).  Why was this group not reported with the others?

3.4:  clarify the direction of the relationship for the information presented at lines 310-316. 

LIne 330:  Boiling was significantly associated with hypertension risk –  this analyses seems to contradict what is presented in Table 2 for boiling.  Perhaps the risk is lowered but still present?

Line 356:  the present study displays a risk with grain consumption and hypertension (Table 1) – which contradicts the statement that these results mirror other results that concluded no significant association between refined grain consumption and hypertension risk.  Where is the data in the current report showing no risk for hypertension with elevated refined grain consumption?  The statement at line 432 adds to the confusion along this topic. 

Line 411: the data are complex but this study does not tease out foods cooked at home versus foods purchased away from home or that the accompanying entrée may actually be the culprit – not the grain itself.  The authors must acknowledge these limitations. 

Round 2

Reviewer 3 Report

Comments and Suggestions for Authors

Thank you for a very well composed response to my comments.  Can you please add a sentence or two to 'methods' to clarify each of the points you make for comments 4, 5, and 6.  This would assure future readers have an understanding of your processes handling data.  
